# Parental Preferences for Expanded Newborn Screening: What Are the Limits?

**DOI:** 10.3390/children10081362

**Published:** 2023-08-09

**Authors:** Nicole S. Y. Liang, Abby Watts-Dickens, David Chitayat, Riyana Babul-Hirji, Pranesh Chakraborty, Robin Z. Hayeems

**Affiliations:** 1Department of Genetic Counselling, The Hospital for Sick Children, Toronto, ON M5G 1X8, Canada; 2Division of Clinical and Metabolic Genetics, The Hospital for Sick Children, Toronto, ON M5G 1X8, Canada; 3Department of Molecular Genetics, University of Toronto, Toronto, ON M5S 1A8, Canada; 4Newborn Screening Ontario, Ottawa, ON K1H 8M8, Canada; 5Child Health Evaluative Sciences, The Hospital for Sick Children, Toronto, ON M5G 1X8, Canada; 6Health Policy, Management and Evaluation, University of Toronto, ON M5T 3M6, Canada

**Keywords:** newborn screening, next-generation sequencing, newborn genomic sequencing

## Abstract

The use of next-generation sequencing technologies such as genomic sequencing in newborn screening (NBS) could enable the detection of a broader range of conditions. We explored parental preferences and attitudes towards screening for conditions for which varying types of treatment exist with a cross-sectional survey completed by 100 parents of newborns who received NBS in Ontario, Canada. The survey included four vignettes illustrative of hypothetical screening targets, followed by questions assessing parental attitudes. Chi-square tests were used to compare frequency distributions of preferences. Results show that most parents supported NBS for conditions for which only supportive interventions are available, but to a significantly lesser degree than those with disease-specific treatments (99% vs. 82–87%, *p* ≤ 0.01). For conditions without an effective treatment, the type of supportive care and age of onset of the condition did not significantly alter parent perceptions of risks and benefits. Parents are interested in expanded NBS for conditions with only supportive interventions in childhood, despite lower levels of perceived benefit for the child and greater anticipated anxiety from screen-positive results. These preferences suggest that the expansion of NBS may require ongoing deliberation of perceived benefits and risks and enhanced approaches to education, consent, and support.

## 1. Introduction

Newborn Screening (NBS) is a highly successful public health program that screens infants for treatable diseases. The purpose of NBS is to detect conditions for which there are effective treatments and provide these treatments to improve health and/or developmental outcomes in infants and children. In Ontario, Canada, Newborn Screening Ontario (NSO) coordinates a provincial program that screens over 140,000 newborns annually for approximately 30 target conditions [1]. Like most NBS programs in North America, NSO primarily utilizes biochemical tests ((e.g., amino acid and acylcarnitine profiling using tandem mass spectrometry (LC-MS/MS)), as “first-tier” tests. Increasingly, targeted molecular tests are also being used as first-tier tests (e.g., qPCR testing for severe combined immune deficiencies, and single nucleotide variant panel testing for *SMN1* and *GJB2* variants associated with spinal muscular atrophy and congenital hearing loss, respectively). More complex biochemical and molecular testing, including next-generation sequencing (NGS), is performed as a “second-tier” test to improve specificity. As the use (NGS) technologies continue to increase in clinical practice [2], it is likely to be integrated into NBS (NGS-NBS) as “first-tier” testing in some jurisdictions [3]. Integration of NGS into NBS would also enable scalable expansion; new tests can be added efficiently and with minimal additional costs [4].

Studies have demonstrated the technical feasibility of NGS-based NBS for conditions with and without available biochemical assays [2,3,5]. This includes both genome-wide sequencing (GWS) and targeted gene panel approaches. For a proposed target disease, evidence supporting the sensitivity of an NGS-based approach is crucial before any consideration of its use as a primary screening test [6,7,8]. There are a number of ongoing research initiatives evaluating the feasibility of genomic NBS to investigate this [9]. In addition, the integration of NGS into NBS enables the detection of conditions that extend beyond current NBS panels [10,11] and challenges the traditional paradigm of NBS focused on the detection of early-onset conditions with effective treatment and aligned with Wilson and Jungner’s screening principles [12,13,14,15].

The emerging literature shows both interest in and concerns about expanded approaches to NBS. For example, DeLuca [16] found that screening for conditions with no treatment was acceptable to the public (84%). Another study showed that 90% of parents definitely wanted to receive results for childhood-onset, medically actionable conditions through NGS-NBS, but the majority (69%) also wanted to receive results for childhood-onset, non-medically actionable conditions [17]. Collectively, these studies show that parents perceive benefits beyond medical treatment in NBS, arguably questioning whether Wilson and Junger’s screening principles need to be adapted to include benefits other than the early initiation of treatment in this context. Support for expanded NBS has also been described in the Canadian public [18]. However, others report concerns about expansion. For example, Grob et al. [19] raised concerns about the creation of a new class of “patients-in-waiting” and the resultant psychological impacts on parent–child bonding and parental distress. NGS-based NBS would also contribute to the burden of detecting and resolving variants of uncertain significance (VUS) [7]. In addition, previous research has already explored parents’ limited absorptive capacity and its impact on psychological adaptation to NBS results, a concern that may be further exacerbated by the use of genomic technologies as a first-tier test [20]. In summary, the literature documenting the perspectives of stakeholders on NGS-based NBS, especially pertaining to the benefits and risks of the expansion of disease targets, is still unresolved, if not at times conflicting.

Despite increasing technological readiness for NGS-based NBS, debate amongst clinicians, laboratory scientists, policymakers, the public, and parents on an acceptable list of target conditions persists [11,17,21]. Our study explores parent attitudes and preferences towards NBS for a range of pediatric target conditions that extend beyond those that meet current screening criteria in Ontario. Specifically, we aimed to determine parent attitudes towards and consent preferences for NBS for conditions with well-established disease modifying treatments, (i.e., aligned with Wilson and Jungner’s screening principles) compared to those with non-medication-based or investigational treatments in childhood.

## 2. Materials and Methods

### 2.1. Design and Setting

A cross-sectional survey study of parents who received NBS for their infants through Newborn Screening Ontario was conducted.

### 2.2. Sample and Recruitment

Eligible participants were parents of children born at the Ottawa Hospital from September 2018 to January 2019 and who participated in NSO’s Laboratory Feasibility Study for NGS-based NBS and agreed to be recontacted for future research. Although NSO’s Laboratory Feasibility Study only recruited the birth mothers of the newborn, this study was open to either parent of the newborn involved in the previous study. A total of 283 parents were eligible.

All eligible participants received an electronic or mailed survey package, depending on the availability of a contact email. Both the electronic and mailed survey packages included a study invitation letter, a link to the parent survey and a unique study ID. The mailed survey package also included a paper version of the survey with a return envelope. Participants were asked to complete and return the survey electronically or by mail, with survey completion constituting consent.

Participants were offered entry into a gift card draw to receive one of ten $25 CAD Amazon gift cards upon completing the survey. Aligned with Dillman’s Tailored Design Method [22], non-responders received up to three mail or email reminders at 4, 8 and 10 weeks after the initial study invitation.

#### 2.2.1. Data Collection

Data were collected between December 2020 and March 2021.The survey content was informed by the literature [18,23] and developed by the study team (Appendix A). The survey contained four sections: (i) demographic characteristics; (ii) awareness of NBS (iii) preferences and attitudes toward different types of screening targets enabled by NGS-NBS and (iv) trust in the healthcare system.

Demographic characteristics including age group, marital status, education level and the type of NBS result received were collected. We assessed participants’ awareness of NBS by asking five yes/no questions. These questions were informed by a previous study assessing NBS education in Ontario [24]. An additional question was related to whether new technology could enable screening for untreatable conditions.

Participants were then presented with four vignettes and asked to answer a series of Likert-scale questions about their preferences and attitudes towards NBS for the described condition. Vignettes are an effective approach for exploring attitudes, values, and beliefs, especially for sensitive topics [25]. The vignettes illustrated four categories of conditions: (i) childhood-onset conditions which have an effective treatment (e.g., Riboflavin transporter deficiency (RTD)), (ii) childhood-onset conditions which have no disease-modifying medication-related treatment but can benefit from behavioral interventions (e.g., Rett syndrome (RETT)), (iii) childhood-onset conditions for which there is no approved disease-modifying treatment but for which there are ongoing clinical trials or emerging treatments (e.g., Duchenne muscular dystrophy (DMD)), (iv) adolescent-adult-onset conditions for which there is no medication-related treatment, but for which there are morbidity reducing surveillance strategies (e.g., Hypertrophic cardiomyopathy (HCM); Table 1). For the purpose of this study, “effective treatment” refers to effective medication-related treatments which are disease modifying and can prevent the progression and complications associated with the condition. Conditions with “no effective treatment” are herein defined as conditions without available medication-related disease-modifying or curative treatments. Lastly, “support” refers to access to supportive services. The vignettes are provided in Appendix A.

Finally, we asked four questions about participants’ trust in the healthcare system, adapted from a validated measure of public trust in health care [26], which explored dimensions like trust in healthcare providers and trust in government policies.

#### 2.2.2. Data Analysis

The preferences and attitudes towards the different categories of conditions were compared. The number of “agree” vs. “other” responses was reported for each preference question and attitude statement. “Agree” included participants who selected “strongly agree” or “agree” and “other” included individuals who answered “disagree”, “strongly disagree”, or “neither agree nor disagree”. Statistical differences between the categories of conditions were assessed using chi-square tests, followed by pairwise comparisons with Bonferroni’s correction. Two-sided *p*-values < 0.05 indicated significance.

**Table 1 children-10-01362-t001:** Categories of Target Conditions in NBS used in vignettes.

Disease Category	Representative Condition	Details about Representative Condition
(i) Childhood-onset conditions which have an effective treatment and align with current NBS criteria	Riboflavin transporter deficiency (RTD; OMIM 211500, 211530, 614707)	Early detection and treatment for RTD can be life-saving or prevent an infant’s dependence on a ventilator [27] Not currently included in NBS in Ontario No biochemical assay for this condition, but it can be detected by NGS
(ii) Childhood-onset conditions which have no medication-related disease-modifying treatment but can benefit from behavioral interventions	Rett Syndrome (RETT; OMIM 312750)	Neurodevelopmental condition where intervention services can minimize comorbidities [28] No emergent treatments No biochemical assay for this condition, but it can be detected by NGS
(iii) Childhood-onset conditions which have no approved disease-modifying medication but have ongoing clinical trials/emerging treatments	Duchenne Muscular Dystrophy (DMD; OMIM 310200)	There are emerging treatments for DMD, like exon-skipping and gene therapies [29] No approved treatments in Ontario/Canada (at time of writing) Can be detected by NGS
(iv) Adolescent–adult-onset conditions with no medication-related treatment, but can benefit from surveillance	Hypertrophic cardiomyopathy (HCM; OMIM 192600)	Typically, adolescent or adult-onset routine surveillance by echocardiography in childhood may prevent the risk of sudden cardiac death [30] Different policy statements currently recommend various ages for offering predictive testing in children, from infancy to age ten in presence of family history No biochemical assay for this condition, can be detected by NGS

## 3. Results

From our initial sample of 283 potential participants, we received 100 completed surveys, yielding a response rate of 36% (100/275), after excluding eight participants with no current address. Socio-demographic data were not ascertained for the non-responders.

Of the 100 participants, the majority were female (98%); 30–39 years of age (74%); married or common-law (94%); living in a large city (66%) (i.e., population ≥ 500,000) and bachelor’s degree educated or above (76%) (Table 2). Most participants had one or two children (79%), and only two participants (2%) previously received a positive NBS result, specifically for congenital adrenal hyperplasia. Ten participants (10%) had immediate family members who underwent genetic testing in the past.

### 3.1. Awareness of NBS

Almost all participants (98%) were able to identify the primary purpose of NBS. Most participants (81%) were also aware that an infant could receive a screen-positive result without being affected with the condition, and 76% understood the concept of uncertain NBS results. Of the total, 98% were aware that parents could decline NBS. Participants were provided with a brief description of NGS-based NBS at the beginning of the survey. Most participants (93%) understood that newer NBS technologies could identify untreatable conditions. Of all the participants, 83% answered four or more of the NBS awareness questions correctly.

### 3.2. Willingness to Participate in NBS for Conditions with No Disease-Modifying Treatment

A total of 73% of participants indicated they wanted their infant screened for all four conditions described in the vignettes. However, the proportion of individuals who wanted their infant screened for an infant-onset condition with an effective treatment like RTD (99%) was significantly greater than the proportion who supported screening for conditions without a medication-related treatment (RETT, DMD, HCM: 87%, 82%, and 85%, respectively; *p* < 0.001) (Figure 1). No significant differences were identified among conditions for which there were no medication-related treatments, despite the availability of supportive interventions and varied ages of onset.

### 3.3. Perceived Benefits and Risks toward NBS for Different Conditions

#### 3.3.1. Perceived Benefit to the Child and Better Support for Families

Almost all participants (99%) agreed that screening for conditions like RTD would be beneficial for children compared to 82% for conditions like RETT (*p* < 0.01), 81% for conditions like DMD (*p* < 0.001) and 77% for conditions like HCM (*p* < 0.0001). When we examined the breakdown of the Likert scale responses, most participants who “agreed” that screening for conditions like RTD would be beneficial for the child selected “strongly agree”, whereas fewer participants selected “strongly agree” for RETT/DMD/HCM. Some participants did not perceive benefit to screening for the described condition but would still want their child screened for it. In addition, the majority of parents agreed that information from NGS-NBS would inform their future pregnancies for RTD (74%), RETT (72%), DMD (78%) and HCM (66%); there were no significant differences between these conditions.

Furthermore, all participants (100%) agreed that screening for conditions like RTD would allow for better support of the child and the family and this agreement was significantly greater compared to conditions like DMD (90%, *p* < 0.05) and HCM (87%, *p* < 0.01), but not compared to conditions like RETT (94%, *p* > 0.05).

#### 3.3.2. Interference with Parent–Child Bonding, Parental Anxiety, and Stigma

One of the key concerns in expanding NBS to include conditions that do not meet current screening criteria relates to the potential disruption in parent–child bonding. Even for conditions like RTD, which would have effective treatment, 14% of participants were concerned that screening would interrupt parent–child bonding compared with 25%, 27%, 13% for conditions like RETT, DMD and HCM, respectively. Parent levels of concern were not equal across the four illustrative conditions (*p* < 0.05), but pairwise comparisons did not reveal significant differences.

Regarding parental anxiety, a lower proportion (11%) of participants anticipated parental anxiety in receiving a true positive result for a condition like RTD compared to 25%, 31% and 34% of participants for conditions like RETT, DMD and HCM, respectively (Figure 2). Participants anticipated significantly greater parental anxiety levels when comparing conditions like DMD (*p* < 0.01) and HCM (*p* < 0.01) with conditions like RTD.

With respect to stigmatization, our results show that only 15% of participants were worried about stigma for their child when screening for a condition like RTD, whereas for conditions like RETT, DMD, and HCM, they anticipated greater worry (31%, 30% and 31%, respectively). The proportions of participants who worried about stigma varied across the four conditions (*p* < 0.05), but pairwise comparisons did not reveal significant differences.

#### 3.3.3. Uncertain Results

Compared with true positive results, a lower proportion of participants agreed that receiving an uncertain result would be beneficial for the child; 69% for conditions like RTD and 57%, 58% and 60% for conditions like RETT, DMD and HCM, respectively. Similarly, a higher proportion agreed that receiving an uncertain result compared with a true positive result would cause unnecessary parental anxiety. This ranged from 47% of participants for conditions like RTD to 54%, 55%, and 56% for conditions like RETT, DMD and HCM. The differences in attitudes towards uncertain results were not statistically significant.

### 3.4. Consent Preferences

A minority (29%) thought that parents should be required to have their baby screened for RTD-like conditions, which have an effective treatment. These numbers are similar for RETT, DMD, and HCM-like conditions (22%, 23% and 21%, respectively). When asked if parents should be strongly encouraged to screen for RTD-like conditions, 91% said yes, significantly more than the 76%, 71% and 74% of participants for conditions like RETT, DMD, and HCM, respectively (Table 3). Lastly, a majority of participants (66%) believed that parents should be able to freely choose, without any guidance from healthcare providers, whether they wanted their child screened for conditions like RTD. For all three representative conditions with no treatment (RETT, DMD, and HCM), 76% agreed with this option.

### 3.5. Trust in the Healthcare System

The majority (85%) of participants agreed that the government is responsible for ensuring a high-quality health care system. Eighty-six percent of participants believed that the government would only fund a medical test if it was worthwhile. Fifty-eight percent of participants believed that the government would not fund a medical test if they were not sure of its benefits.

## 4. Discussion

Our findings reflect on parents’ attitudes and preferences about conditions that could potentially be included in an expanded panel of NBS targets with NGS-based approaches. Parents convey an interest in screening for conditions even if a medication-related treatment is not available, which is one of the key aspects of the traditional Wilson and Junger criterion. This is consistent with studies reporting parental interest in receiving non-medically actionable genome sequencing results for newborns. In fact, the proportion of parents in our study interested (82–87%) is higher than that in the previous studies, where ~70% of parents were interested in receiving genomic results for non-medically actionable conditions in the newborn period [31,32,33], though the definitions of medically actionable varies between studies.

Our findings suggest that the nature of the intervention available and age of onset for conditions without medication-related treatments did not significantly alter parent preferences. Overall, parents perceived greater risks and fewer benefits associated with screening for conditions without medication-related treatments, but these preferences were similar among the three categories of conditions explored. Despite perceived risks, most participants still anticipated benefits to the child and their families and would proceed with screening their child for these conditions if offered.

Much of the discourse for NBS programs rests on medical actionability or availability of effective treatment, which can reduce harm to pre-symptomatic infants. Previous research shows that medical actionability is a major factor in parental decision-making for NBS [17]. At the same time, medical actionability can be regarded as a continuum rather than as a binary phenomenon. There may be different levels of “actionability” that are acceptable to different parents [34]. Milko and colleagues [21] developed an age-based semi-quantitative metric to assess the clinical actionability of returnable results in newborn genomic-sequencing-based severity of disease and the efficacy of the available interventions, the latter of which they subdivided into not, minimally, modestly, and highly effective interventions. The illustrative conditions with no effective disease-specific treatments described in our study (RETT, DMD, HCM) would have interventions that fall into the minimally/no effective categories, yet 77–82% of parents surveyed herein would consider these interventions beneficial to the child. Additionally, 87–94% of parents believed that receiving a true positive result for the representative conditions without effective treatment would lead to better support for the child [35].

NBS for neurodevelopmental syndromes has garnered additional scrutiny regarding the potential for stigmatization and disrupted parent–child bonding. Boardman et al. [35] suggested that the disruption of parent–child bonding might be more of a concern in NBS for conditions predominantly characterized by cognitive and behavioral differences than those predominantly physical in presentation (e.g., HCM, DMD). Our results do not support this; there was no significant difference when comparing parents’ concern for interruption of parent–child bonding for conditions like RETT compared to the other conditions that are predominantly physical in presentation. Neurodevelopmental syndromes such as Rett and Fragile X syndrome are particularly interesting in the NBS context, because even though there is evidence that early intervention can improve cognitive and behavioral outcomes [36,37,38], our ability to diagnose children with these conditions is often fraught with long diagnostic odysseys for families [36,37,38,39,40]. There may be critical developmental periods during which interventions may improve health and developmental outcomes. It is challenging to generate evidence for early intervention strategies because neurodevelopmental conditions are rarely diagnosed pre-symptomatically in the absence of positive family history [28,41]. Thus, if timely diagnosis following symptom onset was possible, NBS may not be necessary.

Parental anxiety following NBS is another concern for expanding the scope of screening. More participants anticipated anxiety in receiving a true positive result for conditions like DMD and HCM compared to conditions like RTD [42]. Surprisingly, there was no significant difference in anticipated parental anxiety between RTD and RETT, suggesting conditions like RETT may be more acceptable for NBS than conditions like DMD and HCM. Two factors may explain this. Firstly, a previous study on genomic sequencing for non-medically actionable conditions showed that parents were more interested in knowing about genetic disorders with more severe manifestations but also anticipated these results to be more distressing [43]. While our results show that parents anticipate more anxiety associated with conditions like DMD where the manifestations are severe, parental interest in results related to conditions like DMD and HCM were similar. Secondly, these differences in anticipated parental anxiety may be attributed to the amount of support parents anticipate after a true-positive result. In fact, significantly fewer parents anticipated improved support for their child and families if they received a true positive result for DMD and HCM. In contrast, all participants anticipated better support for their child and their family for conditions like RTD. There was no significant difference between conditions like RTD and RETT, suggesting that despite the absence of effective treatment for RETT, parents value supportive services like early intervention for their child more than surveillance and the possibility of emerging effective treatments.

As expected, participants indicated that receiving uncertain results from NBS would yield less support and increased parental anxiety compared to receiving true-positive results. Interestingly, there were no meaningful differences in anticipated support and parental anxiety related to uncertain results between conditions with and without medication-related treatments. This suggests that perceived benefits and risks of uncertain results are likely driven by uncertainty itself and not by the characteristics of the condition. We know from previous studies that, compared to parents who received true positive/negative results through NBS, parents who received uncertain results reported more significant uncertainty and struggled with the meaning of an uncertain diagnosis [7,44]. Likewise, uncertain results are also a source of concern for healthcare providers due to a lack of diagnostic and prognostic clarity [45].

For NGS-NBS, uncertainty is likely to increase due to the increased probability of detecting VUS [7]. As seen in other newborn genomic sequencing studies, NBS programs have to make decisions about reporting only pathogenic and likely pathogenic variants according to the accepted practice guidelines developed by the American College of Medical Genetics and Genomics, or also to report VUS. For example, the BabySeq study, which examined genomic sequencing in newborns, only reported pathogenic/likely pathogenic variants unless the enrolled newborn was sick or had specific indications, limiting the reported number of VUS [13]. Moreso, Bodian et al. [2] showed that using an NGS-based approach produced a significantly greater number of uncertain results than a conventional NBS approach (0.90% vs. 0.013%). In current NBS programs where genetic testing is utilized after a screen positive biochemical result, VUS may be reported only for infants with an *a priori* suspicion of the condition [44]. However, if NGS-NBS were to be used as a first-tier test for conditions like RETT and HCM, where there are limited clinical tests to clarify the infant’s phenotype, reporting VUS would place increased demands on NBS reporting systems, families, and health care systems implicated in surveillance for these infants. For example, a longitudinal study by Gonska et al. [46] showed that 10% infants with inconclusive diagnoses of CF due to a VUS in the CFTR gene eventually received a CF diagnosis after the reinterpretation of their VUS. As such, the generation of VUS through NGS-NBS has to be carefully evaluated and monitored to minimize uncertain results and the overmedicalization of infants, but also to identify asymptomatic infants with uncertain genotypes who may benefit from surveillance.

Even for genetic diseases where likely pathogenic and pathogenic variants are detected, disease penetrance may not be well characterized. While these types of results are not VUS, reduced penetrance can create uncertainty for parents in the newborn period [47]. Lewis et al. [43] show that when considering non-medically actionable conditions, parents are more interested in learning about highly penetrant conditions. We did not ask about uncertainty related to penetrance in our study, but it is a dimension that could be explored in future research. In the case of genetic conditions for which there are emerging treatments, like DMD, there is uncertainty in whether a child will have access to clinical trials and research. For such conditions, it is possible to screen specifically for the variants that are eligible for the personalized treatment, like exon-skipping therapy or gene therapy trials in the case of DMD. For DMD specifically, there is also growing evidence that exon-skipping therapies should be provided as early as possible for maximum therapeutic benefits [48].

For conditions with highly effective treatments that can prevent irreversible symptoms in asymptomatic infants, implied consent has been a long-accepted model for NBS programs [6,49,50]. Our results reaffirm that the majority of participants did not agree with mandatory NBS, regardless of the availability of medication-related treatment [18]. Furthermore, our results align with that of previous research suggesting that the expansion of NBS to include conditions with non-medication-related treatments may require an informed consent approach that enables parents to deliberate perceived benefits and harms in screening for such conditions. A significantly greater proportion (91%) of participants agreed that parents should be strongly encouraged to screen for conditions with effective medication-related treatments compared to conditions without medication-related treatment (71–76%). In combination, these results indicate that while participants may be interested in receiving NGS-NBS for conditions without medication-related treatment, they acknowledge that other parents should be able to make a decision that reflects their own values and beliefs. An informed consent process that allows sufficient time for deliberation may also play a role in minimizing potentially negative psychological impacts of screening for conditions without medication-related treatment. To illustrate this, Downie et al. [33] found no difference in levels of anxiety, decisional conflict or regret in parents who had a choice about receiving NBS results for childhood-onset conditions with medical actionability compared to childhood-onset conditions with and without medical actionability. These families consented through the provision of a decision support tool and a genetic counselling session. One proposed solution for the inclusion of conditions that do not align with traditional screening principles, for example, is the use of an implied consent model for conditions that align with “standard” NBS combined with the use of an explicit consent model for late-onset or untreatable conditions [14,50,51,52].

A limitation of our study in terms of generalizability is that this cohort was highly educated and demonstrated high levels of NBS awareness which may not be representative of the general population [24]. The sample size is also relatively small, which may have limited our ability to detect additional differences. Our sampling method was unable to capture the perspective of parents who declined NBS for their child; however, in Ontario, only 0.094% of parents decline NBS. Our sample was also limited in the representation of fathers. Furthermore, our participants were asked to make hypothetical decisions based on vignettes, though research suggests that hypothetical decisions resemble real-world decisions [53]. In addition, our exploration of parental preferences included a limited number of disease categories and attributes. We explored a range of disease categories within a spectrum of treatability but did not provide a category of conditions with truly “no treatment”. This limited our ability to explore important issues like the value of limiting diagnostic odysseys or using results from NBS to inform care decisions (e.g., extubation and palliative care). Other disease attributes which could have been used to delineate categories (e.g., penetrance of a condition, invasiveness of treatment, rate of disease progression) were not explored. Due to our focus on studying a range of disease categories and attributes, important topics related to NGS, such as privacy and insurability, were out of scope. Had respondents been asked to balance these concerns about sequencing technologies with the potential benefits of screening for certain categories of condition, their perspectives may have differed. Future research could examine these attributes to generate insight on parental preferences on a more fulsome spectrum of screening targets enabled by NGS-NBS. In addition, even though we explored consent preferences for screening for conditions without mediation-related treatment, we did not examine practical models of delivering consent.

Our descriptive study showed that parents are interested in expanding NBS for conditions with only supportive interventions in childhood, even with lower levels of anticipated benefit for the child and greater potential for parental anxiety and disruption of parent–child bonding. There were no substantial differences in parental preferences and attitudes when comparing conditions with varying ages of onset and types of interventions. Our data also indicate that regardless of the category of target conditions, the return of uncertain results should be actively minimized. Lastly, there is significantly lower agreement that parents should be strongly encouraged to screen for conditions without medication-related treatment compared to other consent models, even if participants themselves were interested in NGS-NBS for these conditions. The lack of consensus in parental preferences signifies that the expansion of screening targets by NGS-NBS will likely require additional deliberation at the level of screening programs and advisory committees and potential changes to the traditional implied consent model. However, deliberative processes may not need to be tailored to specific conditions. Ongoing research that explores the acceptability of screening for additional target conditions is crucial to informing NGS-based NBS that is consistent with societal values and preferences.

## Figures and Tables

**Figure 1 children-10-01362-f001:**
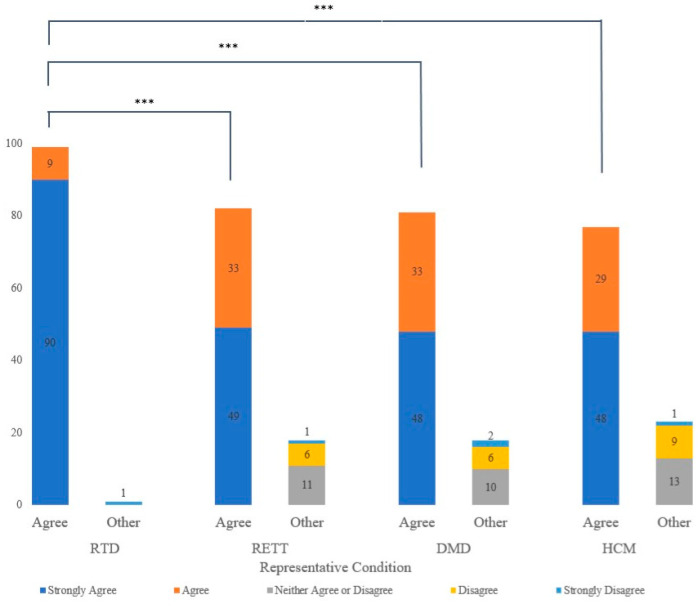
Count of participants who would want their child to undergo NBS for conditions similar to the representative conditions above. RTD = riboflavin transporter deficiency; RETT = Rett syndrome; DMD = Duchenne muscular dystrophy; HCM= Hypertrophic cardiomyopathy. Chi-square between all four representative conditions = *p* < 0.0001. Asterisks indicate statistically significant pairwise comparisons between representative conditions. *** *p* < 0.001.

**Figure 2 children-10-01362-f002:**
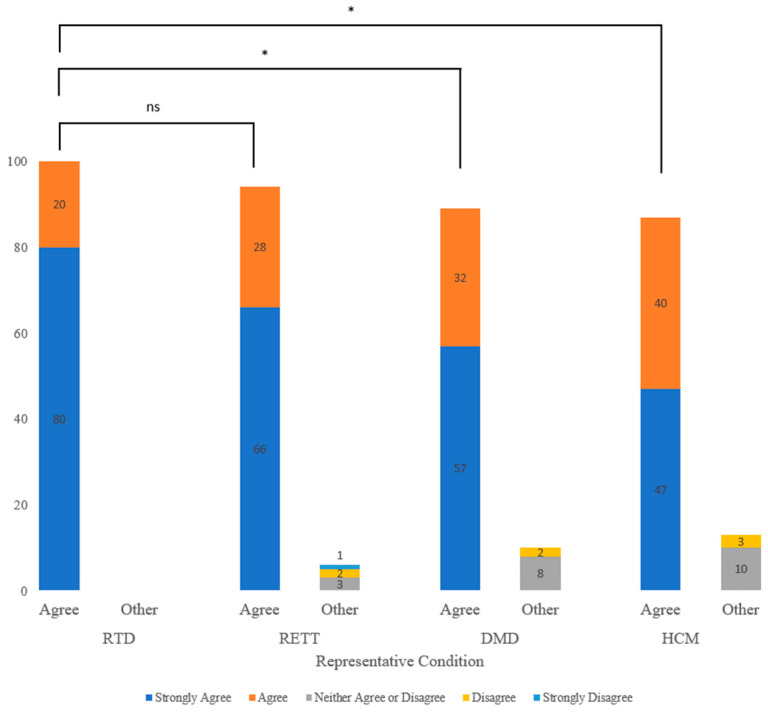
Counts of Participants who “agree” with the attitude statement “Receiving a true positive result from newborn screening for this condition would cause unnecessary parental anxiety.” by condition category. “Agree” includes participants who selected “strongly agree” or “agree”. RTD = riboflavin transporter deficiency; RETT = Rett syndrome; DMD = Duchenne muscular dystrophy; HCM= Hypertrophic cardiomyopathy. Chi-square between all four representative conditions= *p* < 0.0027. Asterisks indicate statistically significant pairwise comparisons between representative conditions. ns = no significance. * < 0.05.

**Table 2 children-10-01362-t002:** Demographic Characteristics of Study Participant.

Characteristic	Total Participants (*n* = 100)
Gender	*n* (%)
Female	98 (98%)
Male	2 (2%)
Age	
20–29	8 (8%)
30–39	74 (74%)
40–49	18 (18%)
Marital status	
Married or living common-law	94 (94%)
Other	6 (6%)
Highest level of education	
High school or less	4 (4%)
College/Trade diploma	20 (20%)
University degree or higher	76 (76%)
Population of city or town	
Rural area	9 (9%)
Small city/town (less than 100,000 people)	12 (12%)
Medium-sized city (100,000–499,999 people)	12 (12%)
Large city (500,000 or more people)	66 (66%)
Unknown	1 (1%)
Number of children	
One	40 (40%)
Two	39 (39%)
Three or more	21 (21%)
Experience with positive NBS results	
Yes	2 (2%)
No	94 (94%)
Do not recall	4 (4%)
Experience with genetic testing in a family member	
Yes	10 (10%)
No	86 (86%)
I do not know	4 (4%)

**Table 3 children-10-01362-t003:** Consent Preferences for Different Categories of Target Conditions.

Q: “For This Condition, Parents Should Be …”	RTD	RETT	DMD	HCM
**… required to have their baby screened.**				
**Yes**	29 (29%)	22 (22%)	23 (24%)	21 (21%)
**No**	71 (71%)	76 (78%)	75 (76%)	79 (79%)
**…strongly encouraged to have their baby screened, but parents can still decline.**				
**Yes**	91 (91%)	76 (76%)	71 (72%)	74 (74%)
**No**	9 (9%)	24 (24%)	28 (28%)	26 (26%)
**…able to choose whether they want their baby screened.**				
**Yes**	66 (66%)	76 (76%)	76 (77%)	76 (76%)
**No**	34 (34%)	24 (24%)	23 (23%)	24 (24%)

## Data Availability

The raw data are available from the authors upon reasonable request.

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
