# Peer review of "Parental Preferences for Expanded Newborn Screening: What Are the Limits?"

_children, 2023, doi:10.3390/children10081362_

Round 1
Reviewer 1 Report
Parental preferences for next generation sequencing-based newborn screening: What are the limits?
In this study the authors explored parental preferences and attitudes towards screening for conditions for which effective treatments do and do not exist with a cross-sectional survey completed by 100 parents of newborns using next-generation sequencing technologies in Ontario, Canada. They found that parents were interested in expanded neonatal screening for conditions with only supportive interventions in childhood, despite lower levels of perceived benefit for the child and greater anticipated anxiety from screen-positive results.
The manuscript is well written. Although the sample size and the proportion of responders are small (this is already addressed in the limitations of the study), it seems methodologically correct.
Minor points.
Figure 1. Three asterisks are used in the figure for comparisons, but only one in the legend. Is that correct? Please, double check it. In addition, for p-values smaller than 0.001, as seem to be the case, “<0.001” should be used, instead of “p<1.547 e-11”, which is less quick to interpret.
Line 186. Again, instead of “(p= 0.00062)”, “p <0.001” is more correct. And more examples through the manuscript…
Figure 2. The significance “(*p<0.0027)” is exactly the same for both comparisons? That seems like a strange coincidence.
Author Response
Response to Reviewer 1 Comments
Point 1 In this study the authors explored parental preferences and attitudes towards screening for conditions for which effective treatments do and do not exist with a cross-sectional survey completed by 100 parents of newborns using next-generation sequencing technologies in Ontario, Canada. They found that parents were interested in expanded neonatal screening for conditions with only supportive interventions in childhood, despite lower levels of perceived benefit for the child and greater anticipated anxiety from screen-positive results.
The manuscript is well written. Although the sample size and the proportion of responders are small (this is already addressed in the limitations of the study), it seems methodologically correct.
Response 1: Thank you for this kind comment.
Point 2: Figure 1. Three asterisks are used in the figure for comparisons, but only one in the legend. Is that correct? Please, double check it. In addition, for p-values smaller than 0.001, as seem to be the case, “<0.001” should be used, instead of “p<1.547 e-11”, which is less quick to interpret.
Line 186. Again, instead of “(p= 0.00062)”, “p <0.001” is more correct. And more examples through the manuscript…
Response 2: Thank you for these comments.
We recognize we did not present our statistical values clearly in our figures. We should have denoted the results of the chi-square analysis, and then subsequently defined the asterisks in the Figure description. Please see the revised Figure descriptions for Figure 1 and Figure 2.
The suggested changes have been made throughout the results section, with all p-values in the format of p < x and/or p > x.
Point 3 : Figure 2. The significance “(*p<0.0027)” is exactly the same for both comparisons? That seems like a strange coincidence.
Response 3: Please see this response in conjuction with Response 2. Our previous figure description did a poor job of representating our statistical analyses. We have presented new figure descriptions on lines 183 and lines 219 which we hope are satisfactory.
Reviewer 2 Report
The authors present an interesting survey study of value to this public health endeavor using original scenarios to facilitate responses on a Likest-type scaled study. Minor revisions are suggested to 1) clarify sequence when comparing percentages and the order referencing); 2) please expand a bit exactly how/why the authors conclude that deliberation is necessary "preferences suggest that the expansion of NBS may require ongoing deliberation of perceived benefits and risks and enhanced approaches to education, consent, and support.", and 3)for improving the readability of the "titles" for all the supplemental figures as they are cumbersome as shown: (see example pasted herein)
Supplementary Figure 4. Counts of Participants who 'Agree’ with the attitude statement "Receiving an uncertain result from newborn screening for this condition would cause unnecessary parental anxiety." by condition category. ‘Agree’ includes participants who selected ‘Strongly Agree’ or ‘Agree’.
This reviewer welcomes any necessary re-re-reads.
Author Response
Response to Reviewer 2 Comments
Point 1 Clarify sequence when comparing percentages and the order referencing
Response 1: We would like to request additional clarification regarding the required revision The authors are not clear on the nature of the revision required.
Point 2: Please expand a bit exactly how/why the authors conclude that deliberation is necessary "preferences suggest that the expansion of NBS may require ongoing deliberation of perceived benefits and risks and enhanced approaches to education, consent, and support."
Response 2: Thank you for this comment. We have revised the final paragraph of the paper (Line 452) to read: “ signifies that the expansion of screening targets by NGS-NBS will likely require additional deliberation at the level of screening programs and advisory committees and potential changes to the traditional implied consent model.”
Point 3 : For improving the readability of the "titles" for all the supplemental figures as they are cumbersome as shown: (see example pasted herein)
Supplementary Figure 4. Counts of Participants who 'Agree’ with the attitude statement "Receiving an uncertain result from newborn screening for this condition would cause unnecessary parental anxiety." by condition category. ‘Agree’ includes participants who selected ‘Strongly Agree’ or ‘Agree’.
Response 3: We agree that the Figure titles are cumbersome and benefit from simplification. Please see the revised Figure titles in our supplementary materials.
Reviewer 3 Report
This study addresses an important question regarding genomic screening of newborns for non-actionable disorders. NBS tenets, usually quoted as being based on general screening guidelines put forth by Wilson and Jungner in the 1960s, propose that disorders must be actionable to qualify for screening. Guidelines derived have also concluded that benefits of avoiding diagnostic odysseys should not be a justification for screening for non-actionable disorder, and there are harms in the early diagnosis of such disorders in robbing parents of enjoyment of time when the child is still well and a problem is not yet suspected. This is an ethically challenging topic, and the authors present data for consideration from a parental perspective that actionability should not preclude such screening, but that parents should have a choice.
The data were generated via a questionnaire on a cohort of subjects that had been enrolled in a prior NBS study.
1. 100 subjects were enrolled from 283 approached (38%) suggesting the potential for bias influencing the results, particularly given the apparent high SES of the responders. Because it is implied that demographic data were already available on the full cohort, could the authors compare characteristics in Table 1 of responders and non-responders and if these are significantly different, address any potential impact in the discussion?
2. It is this reviewer’s opinion that the survey is flawed by not presenting a scenario of true non-actionability. Scenarios 2, 3 and 4 all imply the involvement of some kind of intervention which although not curative, may still provide benefit to the child and parents. For #2, behavioral interventions that decrease comorbidities (RETT) are a benefit. For #3 (DMD), early initiation of steroids have been shown to lengthen time that patients can walk and improve quality of life – both benefits. Survellance in #4, while not a “treatment”, may definitely be important to preventing morbidity and mortality. Missing, as a scenario, is an example of a disorder for which there is absolutely no intervention that is beneficial (beyond providing a diagnosis to the parents so that they understand what the problem is and can plan around that, and the minimization of diagnostic odysseys). It is conceivable to consider options 2-4 interventions as equivalent to therapy, given that outcomes may be improved. Thus the questionnaire is really not surveying opinion on “no treatment” in the sense of no intervention that may be of value to the child and parents. While the investigators may not be able to go back and collect these data, they could still present valuable information on parental perspective by carefully redefining “no treatment” (e.g. as no drug or diet change that must be initiated immediately to impact outcome) and expanding the definition of what they asked about to a spectrum of interventions ranging from initiating immediate drug therapy to initiating a surveillance protocol, and discussing the fine points between NBS use for disorders with a therapy vs disorders where outcomes may still be changed but with institution of non-pharmacologic early interventions, including behavioral and surveillance that can improve outcome.
3. In the supplementary material that includes the actual questionnaire, it is clear to me that not enough is explained about next generation sequencing (NGS) given that the technology is just described briefly as the use of DNA. The word “sequencing” was not included. There is an entire literature on the fears of people about DNA sequencing with regards to invasion of privacy, how information is stored and who has access, and insurability, as well as how the information must be handled when a minor surpasses 18 years of age (i.e. if NGS is generating results on sequence variants, do they need to be retained and consent obtained for rerelease at a later age?). The title of the paper includes the term NGS, and yet the questionnaire does not address any of the baggage that goes along with sequencing outside of the issues of screening for non-actionable disorders.
4. P. 6 line 168: The use fo the term “no effective treatment”, as discussed above in #2, is really not appropriate, as there are interventions that impact outcome for these disorders.
5. For the figures, it might be easier for the reader to interpret the Likert scale categories with patterns rather than what appears as shades of gray in the copy that I printed. The shades of gray har hard to distinguish on a black and white version of a printout.
6. 3.32 p.6 line 199: Paragraph on Interference with parent-child bonding, parental anxiety and stigma. Pereira S, Smith HS, Frankel LA, Christensen KD, Islam R, Robinson JO, Genetti CA, Blout Zawatsky CL, Zettler B, Parad RB, Waisbren SE, Beggs AH, Green RC, Holm IA, McGuire AL; BabySeq Project Team. Psychosocial Effect of Newborn Genomic Sequencing on Families in the BabySeq Project: A Randomized Clinical Trial. JAMA Pediatr. 2021 Nov 1;175(11):1132-1141. doi: 10.1001/jamapediatrics.2021.2829. PMID: 34424265; PMCID: PMC8383160. This reference is not included, but would be appropriate to consider given it provides data showing a lack of impact of NGS results on parental bonding and anxiety.
7. While the reviewer is struck by the consistently and surprisingly strong response of parents about proceeding with NBS for disorders, it is not possible to know whether the responses might have been entirely different if the questionnaire’s background information was presented differently (e.g. no treatment meaning no intervention, use of NGS instead of a vague DNA reference).
8. I am surprised the investigators did not also ask questions about parental views on avoidance of diagnostic odyssey, which is often considered to not be a reason to justify NBS, but falls into the spectrum of screening for disorders with “no treatment”.
9. With modifications suggested above, I believe the information is important, as it suggests a difference between parental preference and public health newborn screening priorities. Newborn screeners do not want to approach non-actionable disorders outside Wilson and Jungner criteria, yet parents seem to feel that some of the types of disorders being excluded from screening should be included.
Author Response
Response to Reviewer 3 Comments
Point 1 100 subjects were enrolled from 283 approached (38%) suggesting the potential for bias influencing the results, particularly given the apparent high SES of the responders. Because it is implied that demographic data were already available on the full cohort, could the authors compare characteristics in Table 1 of responders and non-responders and if these are significantly different, address any potential impact in the discussion?
Response 1: Thank you for this comment. Demographics data was collected as part of the study survey and therefore, demographics data is unavailable to our study team for non-responders, though this would certainly be a very interesting comparison.
Point 2: It is this reviewer’s opinion that the survey is flawed by not presenting a scenario of true non-actionability. Scenarios 2, 3 and 4 all imply the involvement of some kind of intervention which although not curative, may still provide benefit to the child and parents. For #2, behavioral interventions that decrease comorbidities (RETT) are a benefit. For #3 (DMD), early initiation of steroids have been shown to lengthen time that patients can walk and improve quality of life – both benefits. Survellance in #4, while not a “treatment”, may definitely be important to preventing morbidity and mortality. Missing, as a scenario, is an example of a disorder for which there is absolutely no intervention that is beneficial (beyond providing a diagnosis to the parents so that they understand what the problem is and can plan around that, and the minimization of diagnostic odysseys). It is conceivable to consider options 2-4 interventions as equivalent to therapy, given that outcomes may be improved. Thus the questionnaire is really not surveying opinion on “no treatment” in the sense of no intervention that may be of value to the child and parents. While the investigators may not be able to go back and collect these data, they could still present valuable information on parental perspective by carefully redefining “no treatment” (e.g. as no drug or diet change that must be initiated immediately to impact outcome) and expanding the definition of what they asked about to a spectrum of interventions ranging from initiating immediate drug therapy to initiating a surveillance protocol, and discussing the fine points between NBS use for disorders with a therapy vs disorders where outcomes may still be changed but with institution of non-pharmacologic early interventions, including behavioral and surveillance that can improve outcome.
Response 2: Thank you for this comment on our study questionnare design. We agree that we presented a spectrum of treatability and interventions with Scenarios 2, 3 and 4 and did not present a condition on the far end of the spectrum with no possible treatment or intervention. At the time of survey design,we focussed on a spectrum of treatability rather than on the presence vs absence of treatment as a way to begin to test the boundaries of the Wilson and Junger’s screening principles.
As such, as per your suggestion, we are defining ‘no treatment’ directly in our main text file (line 133) and also acknowledge this study flaw in our limitations (line 427). We regretably are unable to revise our study survey and provide the same definition within the survey.
Point 3 : In the supplementary material that includes the actual questionnaire, it is clear to me that not enough is explained about next generation sequencing (NGS) given that the technology is just described briefly as the use of DNA. The word “sequencing” was not included. There is an entire literature on the fears of people about DNA sequencing with regards to invasion of privacy, how information is stored and who has access, and insurability, as well as how the information must be handled when a minor surpasses 18 years of age (i.e. if NGS is generating results on sequence variants, do they need to be retained and consent obtained for rerelease at a later age?). The title of the paper includes the term NGS, and yet the questionnaire does not address any of the baggage that goes along with sequencing outside of the issues of screening for non-actionable disorders.
Response 3: Thank you for this comment. With the goal of targeting a very specific nuance of NGS use in the context of NBS expansion (types of newborn screening targets), and to avoid an unnecessarily complex set of survey questions, our study survey did not address the full range of concerns of using NGS for public health programs like newborn screening such as privacy, information storage, insurability, and others. We have reflected this in our study limitations (line 433).
Point 4 P. 6 line 168: The use fo the term “no effective treatment”, as discussed above in #2, is really not appropriate, as there are interventions that impact outcome for these disorders.
Response 4:
We feel that our revisions in response to Point 2 alleviates some of the appropriateness of using the term ‘no effective treatment’ as it is now defined for the purposes of our study.
Point 5 For the figures, it might be easier for the reader to interpret the Likert scale categories with patterns rather than what appears as shades of gray in the copy that I printed. The shades of gray har hard to distinguish on a black and white version of a printout.
Response 5: We agree with this comment and a similar suggestion was also made by the academic editor of Children. We have revised the Figure to include figure with colour for easier interpretations.
Point 6 3.32 p.6 line 199: Paragraph on Interference with parent-child bonding, parental anxiety and stigma. Pereira S, Smith HS, Frankel LA, Christensen KD, Islam R, Robinson JO, Genetti CA, Blout Zawatsky CL, Zettler B, Parad RB, Waisbren SE, Beggs AH, Green RC, Holm IA, McGuire AL; BabySeq Project Team. Psychosocial Effect of Newborn Genomic Sequencing on Families in the BabySeq Project: A Randomized Clinical Trial. JAMA Pediatr. 2021 Nov 1;175(11):1132-1141. doi: 10.1001/jamapediatrics.2021.2829. PMID: 34424265; PMCID: PMC8383160. This reference is not included, but would be appropriate to consider given it provides data showing a lack of impact of NGS results on parental bonding and anxiety.
Response 6: This reference has been added to line 330. Thank you for this suggestion.
Point 7 While the reviewer is struck by the consistently and surprisingly strong response of parents about proceeding with NBS for disorders, it is not possible to know whether the responses might have been entirely different if the questionnaire’s background information was presented differently (e.g. no treatment meaning no intervention, use of NGS instead of a vague DNA reference).
Response 7: We agree that our study is limited in this respect and have acknowledged in the Limitations section that the scope of our work was restricted to disease-target preferences (line 433).
Point 8 I am surprised the investigators did not also ask questions about parental views on avoidance of diagnostic odyssey, which is often considered to not be a reason to justify NBS, but falls into the spectrum of screening for disorders with “no treatment”.
Response 8: In responding to and making revisions to previous comments (Points 2 & 4) , we have attempted to clarify that our definition of “effective treatment” was limited to disease-modifying and highly effective treatments which significantly alter disease progression. In making these revisions, we acknowledge that we do not have a vignette which represents condtions where treatments relate to symptom management or do not exist at all. In the absence of a vignette of this nature, the screening justification related to avoiding the diagnostic odyssey is not relevant.
Point 9 With modifications suggested above, I believe the information is important, as it suggests a difference between parental preference and public health newborn screening priorities. Newborn screeners do not want to approach non-actionable disorders outside Wilson and Jungner criteria, yet parents seem to feel that some of the types of disorders being excluded from screening should be included.
Response 9: Thank you for this comment. We agree that this is challenging tension and we hope our evidence can inform NBS decisions going forwards.
Reviewer 4 Report
I am impressed by the tremendous amount of work that the authors put into preparing the study and drafting the manuscript. The assumptions of the work were properly defined. The introduction addresses the basic problems and issues and justifies the purpose of the study. I have no objections to the methodology. The results were discussed in comparison with the relevant literature. Conclusions are based on the results obtained. Particularly noteworthy is the discussion indicating the usefulness of continuing the conducted research on subsequent groups.
Author Response
Response to Reviewer 4 Comments
Point 1 I am impressed by the tremendous amount of work that the authors put into preparing the study and drafting the manuscript. The assumptions of the work were properly defined. The introduction addresses the basic problems and issues and justifies the purpose of the study. I have no objections to the methodology. The results were discussed in comparison with the relevant literature. Conclusions are based on the results obtained. Particularly noteworthy is the discussion indicating the usefulness of continuing the conducted research on subsequent groups.
Response 1: We would ike to thank reviewer for these kind comments and support.
Reviewer 5 Report
Dear authors,
In this study, you explored parents' preferences and attitudes toward screening for conditions for which effective treatments do and do not exist through a cross-sectional survey.
Introduction:
The introduction is well organized and presents relevant information for the content of the article.
Materials and Methods: are well structured, with an adequate description of study design.
Results and discussion:
As you mentioned, the study had limitations. The sample size is relatively small (100 participants), the majority were female , a high percentage of participants (76%) had a bachelor's degree with higher education, and participants were asked to make hypothetical decisions (only 2% had positive NBS results and only 10% had experience with genetic testing for a family member).
In my opinion, the results and discussions should be restructured and shortened and the article requires drawing some conclusions regarding the importance of conducting this study.
Finally I recommend accepting after major revision .
Author Response
Response to Reviewer 5 Comments
Point 1 As you mentioned, the study had limitations. The sample size is relatively small (100 participants), the majority were female , a high percentage of participants (76%) had a bachelor's degree with higher education, and participants were asked to make hypothetical decisions (only 2% had positive NBS results and only 10% had experience with genetic testing for a family member).
In my opinion, the results and discussions should be restructured and shortened and the article requires drawing some conclusions regarding the importance of conducting this study.
Response 1: Thank you for this comment. We are unclear on how the reviewer would like the results and discussion reconstructed. We have removed one section of the Results (i.e. “trust in the health care system” as one content reduction strategy. Otherwise, our preference is to retain the detail currently provided in the body of the discussion.
Round 2
Reviewer 3 Report
Comments on Author’s responses to Reviewer 3
Point 1 100 subjects were enrolled from 283 approached (38%) suggesting the potential for bias influencing the results, particularly given the apparent high SES of the responders. Because it is implied that demographic data were already available on the full cohort, could the authors compare characteristics in Table 1 of responders and non-responders and if these are significantly different, address any potential impact in the discussion?
Response 1: Thank you for this comment. Demographics data was collected as part of the study survey and therefore, demographics data is unavailable to our study team for non-responders, though this would certainly be a very interesting comparison.
Response to Response 1: The manuscript implies that the 100 subjects are a subcohort of a separate feasibility study. The reviewer was unable to find a reference to the original cohort, although perhaps there have been no studies published? Are the authors certain that there are no demographic data on the subjects from the original “Feasibility” study that would not be able to supply any comparators as proxies for SES status? If not, the 38% uptake should be more directly addressed in the discussion of weaknesses, as well as that the skewed SES nature of the cohort may impact results and conslusions.
Point 2: It is this reviewer’s opinion that the survey is flawed by not presenting a scenario of true non-actionability. Scenarios 2, 3 and 4 all imply the involvement of some kind of intervention which although not curative, may still provide benefit to the child and parents. For #2, behavioral interventions that decrease comorbidities (RETT) are a benefit. For #3 (DMD), early initiation of steroids have been shown to lengthen time that patients can walk and improve quality of life – both benefits. Surveillance in #4, while not a “treatment”, may definitely be important to preventing morbidity and mortality. Missing, as a scenario, is an example of a disorder for which there is absolutely no intervention that is beneficial (beyond providing a diagnosis to the parents so that they understand what the problem is and can plan around that, and the minimization of diagnostic odysseys). It is conceivable to consider options 2-4 interventions as equivalent to therapy, given that outcomes may be improved. Thus, the questionnaire is really not surveying opinion on “no treatment” in the sense of no intervention that may be of value to the child and parents. While the investigators may not be able to go back and collect these data, they could still present valuable information on parental perspective by carefully redefining “no treatment” (e.g. as no drug or diet change that must be initiated immediately to impact outcome) and expanding the definition of what they asked about to a spectrum of interventions ranging from initiating immediate drug therapy to initiating a surveillance protocol, and discussing the fine points between NBS use for disorders with a therapy vs disorders where outcomes may still be changed but with institution of non-pharmacologic early interventions, including behavioral and surveillance that can improve outcome.
Response 2: Thank you for this comment on our study questionnare design. We agree that we presented a spectrum of treatability and interventions with Scenarios 2, 3 and 4 and did not present a condition on the far end of the spectrum with no possible treatment or intervention. At the time of survey design,we focussed on a spectrum of treatability rather than on the presence vs absence of treatment as a way to begin to test the boundaries of the Wilson and Junger’s screening principles.
As such, as per your suggestion, we are defining ‘no treatment’ directly in our main text file (line 133) and also acknowledge this study flaw in our limitations (line 427). We regretably are unable to revise our study survey and provide the same definition within the survey.
Response to Response 2: This remains a serious flaw, as perhaps parents would have answered differently if the survey was framed differently.
Point 3 : In the supplementary material that includes the actual questionnaire, it is clear to me that not enough is explained about next generation sequencing (NGS) given that the technology is just described briefly as the use of DNA. The word “sequencing” was not included. There is an entire literature on the fears of people about DNA sequencing with regards to invasion of privacy, how information is stored and who has access, and insurability, as well as how the information must be handled when a minor surpasses 18 years of age (i.e. if NGS is generating results on sequence variants, do they need to be retained and consent obtained for rerelease at a later age?). The title of the paper includes the term NGS, and yet the questionnaire does not address any of the baggage that goes along with sequencing outside of the issues of screening for non-actionable disorders.
Response 3: Thank you for this comment. With the goal of targeting a very specific nuance of NGS use in the context of NBS expansion (types of newborn screening targets), and to avoid an unnecessarily complex set of survey questions, our study survey did not address the full range of concerns of using NGS for public health programs like newborn screening such as privacy, information storage, insurability, and others. We have reflected this in our study limitations (line 433).
Response to Response 3: My feelings are quite strong that the title “Parental preferences for next generation sequencing-based newborn screening: What are the limits?” is inappropriate for this project, given that parents were not specifically asked about NGS based NBS, but rather a vague DNA reference. Given that the authors were actually exploring how parents felt about screening for disorders that might be considered outside of the traditional Wilon and Jungner criterion, that a disorder must have an available treatment to be considered for screening, the title should really be changed. It is understood that the purpose of asking these questions is to get at how parents would feel about identifying disorders through NGS based NBS that are not currently identified, but this is not the framework or spirit in which the questions are presented in the absence of detailed explanation of this issue. Also, given the absence of scenario that included a truly non-treatable disorder, fatal errors are stacking up, unless the authors make it clear to the reader that the results address queries on a spectrum of treatability (exclusive of non-treatability) to explore parents views relative to a most strict interpretation of a Wilson-Jungner treatable disorder, and that the reason for asking these questions is to indirectly apply those results to a world in which NGS based NBS (which might identify non-treatable and just mildly modifiable outcome disorders.
Author Response
Response to Reviewer 3 Comments
*please note that previous comments and revisions for reference. Our most recent responses are in red*
Point 1 100 subjects were enrolled from 283 approached (38%) suggesting the potential for bias influencing the results, particularly given the apparent high SES of the responders. Because it is implied that demographic data were already available on the full cohort, could the authors compare characteristics in Table 1 of responders and non-responders and if these are significantly different, address any potential impact in the discussion?
Response 1: Thank you this comment. Demographics data was collected as part of the study survey and therefore, demographics data is unavailable to our study team for non-responders, though this would certainly be a very interesting comparison.
Response to Response 1: The manuscript implies that the 100 subjects are a subcohort of a separate feasibility study. The reviewer was unable to find a reference to the original cohort, although perhaps there have been no studies published? Are the authors certain that there are no demographic data on the subjects from the original “Feasibility” study that would not be able to supply any comparators as proxies for SES status? If not, the 38% uptake should be more directly addressed in the discussion of weaknesses, as well as that the skewed SES nature of the cohort may impact results and conslusions.
Response #2 to Response 1:
We have reached out to the primary authors of the feasibility study to clarify details on the study subjects. The feasibility study (technical feasibility of performing NGS on dried blood spots in Ontario) has not been published yet but is in progress. Unfortunately, the feasibility study did not collect participant characteristics that align with those collected in the study reported herein. As such, we are unable to report on the alignment between the sudy samples with respect to sociodemographic characteristics. We have added line 158 for clarification.
Point 2: It is this reviewer’s opinion that the survey is flawed by not presenting a scenario of true non-actionability. Scenarios 2, 3 and 4 all imply the involvement of some kind of intervention which although not curative, may still provide benefit to the child and parents. For #2, behavioral interventions that decrease comorbidities (RETT) are a benefit. For #3 (DMD), early initiation of steroids have been shown to lengthen time that patients can walk and improve quality of life – both benefits. Surveillance in #4, while not a “treatment”, may definitely be important to preventing morbidity and mortality. Missing, as a scenario, is an example of a disorder for which there is absolutely no intervention that is beneficial (beyond providing a diagnosis to the parents so that they understand what the problem is and can plan around that, and the minimization of diagnostic odysseys). It is conceivable to consider options 2-4 interventions as equivalent to therapy, given that outcomes may be improved. Thus, the questionnaire is really not surveying opinion on “no treatment” in the sense of no intervention that may be of value to the child and parents. While the investigators may not be able to go back and collect these data, they could still present valuable information on parental perspective by carefully redefining “no treatment” (e.g. as no drug or diet change that must be initiated immediately to impact outcome) and expanding the definition of what they asked about to a spectrum of interventions ranging from initiating immediate drug therapy to initiating a surveillance protocol, and discussing the fine points between NBS use for disorders with a therapy vs disorders where outcomes may still be changed but with institution of non-pharmacologic early interventions, including behavioral and surveillance that can improve outcome.
Response 2: Thank you for this comment on our study questionnaire design. We agree that we presented a spectrum of treatability and interventions with Scenarios 2, 3 and 4 and did not present a condition on the far end of the spectrum with no possible treatment or intervention. At the time of survey design,we focussed on a spectrum of treatability rather than on the presence vs absence of treatment as a way to begin to test the boundaries of the Wilson and Junger’s screening principles.
As such, as per your suggestion, we are defining ‘no treatment’ directly in our main text file (line 133) and also acknowledge this study flaw in our limitations (line 427). We regretfully are unable to revise our study survey and provide the same definition within the survey.
Response to Response 2: This remains a serious flaw, as perhaps parents would have answered differently if the survey was framed differently.
Response #2 to Response 2.
We have made further revisions to better acknowledge the absence of a vignette with no treatment (lines 18, 86, 87, 127, 129, 131, 185, 187, 289, 298, 364, 409, 411, 415, 418, 422 and Table 1). We understand that this remains a study flaw and have tried to modify all areas indicating ‘no treatment’ in our manuscript. We thank you for your thoughtful comment and welcome further suggestions for edits to improve this flaw.
Point 3 : In the supplementary material that includes the actual questionnaire, it is clear to me that not enough is explained about next generation sequencing (NGS) given that the technology is just described briefly as the use of DNA. The word “sequencing” was not included. There is an entire literature on the fears of people about DNA sequencing with regards to invasion of privacy, how information is stored and who has access, and insurability, as well as how the information must be handled when a minor surpasses 18 years of age (i.e. if NGS is generating results on sequence variants, do they need to be retained and consent obtained for rerelease at a later age?). The title of the paper includes the term NGS, and yet the questionnaire does not address any of the baggage that goes along with sequencing outside of the issues of screening for non-actionable disorders.
Response 3: We appreciate that in hopes to targetting a very specific nuance of using NGS in the expansion of NBS (types of newborn screening targets) and to not present a more complex set of survey questions, our study survey did address the full range of concerns of using NGS for public health programs like newborn screening like concerns with privacy, information storage, insurability, amongst other concerns. We have reflected this in our study limitations (line 433).
Response to Response 3: My feelings are quite strong that the title “Parental preferences for next generation sequencing-based newborn screening: What are the limits?” is inappropriate for this project, given that parents were not specifically asked about NGS based NBS, but rather a vague DNA reference. Given that the authors were actually exploring how parents felt about screening for disorders that might be considered outside of the traditional Wilon and Jungner criterion, that a disorder must have an available treatment to be considered for screening, the title should really be changed. It is understood that the purpose of asking these questions is to get at how parents would feel about identifying disorders through NGS based NBS that are not currently identified, but this is not the framework or spirit in which the questions are presented in the absence of detailed explanation of this issue. Also, given the absence of scenario that included a truly non-treatable disorder, fatal errors are stacking up, unless the authors make it clear to the reader that the results address queries on a spectrum of treatability (exclusive of non-treatability) to explore parents views relative to a most strict interpretation of a Wilson-Jungner treatable disorder, and that the reason for asking these questions is to indirectly apply those results to a world in which NGS based NBS (which might identify non-treatable and just mildly modifiable outcome disorders.
Response #2 to Response 3: Thank you for outlining your concerns clearly for us. As suggested, we have changed the title of our paper and have made modifications to line 86.
Reviewer 5 Report
Dear authors, thank you for the changes made and I believe that the article can be accepted for publication in this form.
Author Response
This reviewer's comment did not necessitate an additional response or revision.
Thank you again for the previous comment.